# Effect of Zbed6 Single-Allele Knockout on the Growth and Development of Skeletal Muscle in Mice

**DOI:** 10.3390/biology12020325

**Published:** 2023-02-17

**Authors:** Ling Liu, Shengnan Wang, Wenjie Tian, Cheng Xu, Chengjie Wei, Kai Cui, Lin Jiang, Dandan Wang

**Affiliations:** 1National Germplasm Center of Domestic Animal Resources, Institute of Animal Sciences, Chinese Academy of Agricultural Sciences (CAAS), Beijing 100193, China; 2Key Laboratory of Livestock and Poultry Resources Evaluation and Utilization, Institute of Animal Sciences, Chinese Academy of Agricultural Sciences (CAAS), Beijing 100193, China; 3College of Animal Science and Technology, Qingdao Agricultural University, Qingdao 266109, China

**Keywords:** ZBED6, single-allele knockout, IGF2, BARX2, mice, skeletal muscle

## Abstract

**Simple Summary:**

The growth and development of skeletal muscle influences the efficiency of animal production. ZBED6 knockout accelerated the growth and development of pig and mice by regulating its target gene *Igf2*. However, the effect of, as well as detailed information about, the molecular mechanism of *Zbed6* single-allele knockout on regulating muscle growth and development is still limited. In this study, a model of *Zbed6* single-allele knockout mice prepared by gene-editing technology was used to study the effect and molecular mechanism of *Zbed6* single-allele knockout on mice. Phenotypes and the RNA-seq results of muscle from *Zbed6* single-allele knockout and wild-type mice showed that *Zbed6* single-allele knockout promoted skeletal muscle weight and muscle fiber area via another target gene, instead of *Igf2*. This study may help to further explore the function of ZBED6 and its other target genes that have not been studied.

**Abstract:**

ZBED6, a key transcription factor, plays an important role in skeletal muscle and organ growth. ZBED6 knockout (*ZBED6*^−/−^) leads to the upregulation of *IGF2* in pig and mice muscle, thereby increasing muscle mass. However, the effects and mechanism of *Zbed6* single-allele knockout (*Zbed6*^+/−^) on mice muscle remain unknown. Here, we reported that *Zbed6*^+/−^ promotes muscle growth by a new potential target gene rather than *Igf2* in mice muscle. *Zbed6*^+/−^ mice showed markedly higher muscle mass (25%) and a markedly higher muscle weight ratio (18%) than wild-type (WT) mice, coinciding with a larger muscle fiber area (28%). Despite a significant increase in muscle growth, *Zbed6*^+/−^ mice showed similar *Igf2* expression with WT mice, indicating that a ZBED6–*Igf2*-independent regulatory pathway exists in *Zbed6*^+/−^ mice muscle. RNA-seq of muscle between the *Zbed6*^+/−^ and WT mice revealed two terms related to muscle growth. Overlapping the DEGs and C2C12 Chip-seq data of ZBED6 screened out a potential ZBED6 target gene *Barx2*, which may regulate muscle growth in *Zbed6*^+/−^ mice. These results may open new research directions leading to a better understanding of the integral functions of ZBED6 and provide evidence of *Zbed6*^+/−^ promoting muscle growth by regulating *Barx2* in mice.

## 1. Introduction

Zinc finger BED domain-containing protein 6 (ZBED6), a transcription factor, was discovered as a repressor of *IGF2* (insulin-like growth factor 2) [1]. It is derived and evolved from a domesticated DNA transposon that plays an important role distinctively in placental mammals [2,3]. This gene is encoded by an intronless gene, located in intron 1 of Zc3h11a, and shows a high degree of conservatism in 26 kinds of species with an amino acid identity close to 100%, which suggests the significance of ZBED6 in animal growth and development [1]. 

In the earliest time, a quantitative trait locus (QTL) was found to significantly affect pig muscle growth, fat deposition, and heart size during a conventional breeding process that was located in the *IGF2* gene [4,5]. The mutant of a single base in *IGF2* intron 3 disrupted a binding site of an unknown nucleic factor to *IGF2* (G > A), resulting in a three-fold upregulation of *IGF2* mRNA expression in pig muscle, eventually leading to the increase in muscle mass [6]. The unknown nucleic factor was named ZBED6 and was identified to be a transcription factor that regulates the expression of *Igf2* by binding with the “GCTCG” motif sequence, thereby regulating the growth and development of skeletal muscle [1]. 

Since then, multiple studies have extended the approach to several mammalians and cell lines. In beef cattle, it was found that the three SNPs in the promoter region of the *ZBED6* gene were in complete linkage disequilibrium (LD) in *IGF2*, which revealed the significant effect of ZBED6 on cattle growth traits [7,8]. A study confirmed that *Zbed6* knockout mice and *Igf2* knockin mice destroyed the binding site of ZBED6 and *Igf2*, which simultaneously resulted a in faster growth speed, an increased serum IGF2 level, and increased skeletal muscle mass [2]. These results were also proven in either ZBED6 knockout or *IGF2* intro 3 mutant pigs and revealed the increased effects of ZBED6 knockout on other internal organs [9,10,11,12]. As for specific cells, ZBED6 could not only regulate the proliferation and differentiation of mouse C2C12 cells [13,14] but also regulate other cell functions, including insulin production of mouse MIN6 cells [15], pancreas β Cell proliferation and death [16], proliferation and differentiation of mouse fat precursor cells [17], and regulation of human colorectal cancer cell cycle and growth [18]. In addition, about 2500 binding sites of ZBED6 were identified in mice by ChIP sequencing [1], and some genes other than *IGF2* were proven to be the target genes of ZBED6 that regulate the muscle growth of pigs together [10]. 

In a previous study by Younis et al. [2], the knockout of *Zbed6* increased the growth and development of skeletal muscle in mice. However, the effect exists on *Zbed6*^+/−^ mice is unknown not been proved in single knockout puberty and adult mice till now. Thus, it is important for a single-allele strategy to be performed at both the living and transcriptome levels, to explore a more sophisticated mechanism and regulation network for ZBED6. The only report related to single-allele knockout of *ZBED6* reveals that transcriptome differences vary from different organs in pigs [19]. There are not any research results about whether single-allele knockout causes significant changes in skeletal muscle. More phenotypic and transcriptomic information is needed to fill in the gap in single knockout genotypes and complete the regulation network of ZBED6. In this study, *Zbed6*^+/−^ and wild-type mice (WT) were characterized in littermates and then slaughtered at the age of 8 weeks for muscle dissection to explore the effect of *Zbed6*^+/−^ in mice skeletal muscle, and the muscle traits of two groups were also calculated. RNA-seq analysis of skeletal muscle was performed to confirm the transcriptomic differences between *Zbed6*^+/−^ and WT mice in females to identify some of the ZBED6 target genes functioning in *Zbed6*^+/−^ skeletal muscle development.

## 2. Materials and Methods

### 2.1. Animal Models

The *Zbed6*^+/−^ mouse models in C57BL/6 were provided by Leif Andersson (Science for Life Laboratory, Department of Medical Biochemistry and Microbiology, Uppsala University). The removal of the whole *Zbed6* gene was based on DNA homologous recombination aroused by the Cre-lox system [2]. The mice and their offspring were kept in the Small Animal Experiment Center of the Institute of Animal Science, Chinese Academy of Agricultural Sciences (CAAS), Beijing, China. We obtained WT, *Zbed6*^+/−^, and *Zbed6*^−/−^ mice simultaneously in littermates using *Zbed6*^+/−^ mice crossing. Different kinds of genotype classes of mice were segregated by littermates, and the WT mice were seen as control. The body weight of WT and *Zbed6*^+/−^ female littermates was recorded at the age of 8 weeks. All mice were housed in standard conditions with water and food given ad libitum in the Animal Experiment Center of CAAS. 

### 2.2. Mice Genotyping

All mice were labeled with stainless steel ear marks, and 1–3 mm^3^ tails were collected at the age of 4 weeks to extract genomic DNA using KAPA Express Extract (KK7102, ROCHE). The tail samples were immersed in the mixture of Sup. Protease and Adv. Buffer and then digested for 15 min at 75 °C. Genotyping was performed by PCR and gel electrophoresis. Three oligo-nucleotides were used to detect WT and *Zbed6*^+/−^ simultaneously. The expected product size is 596 bp for WT. *Zbed6*^+/−^ has two products with lengths of 596 and 561 bp. The PCR amplification protocol included pre-denaturation at 95 °C for 10 min, 35 cycles of denaturation at 95 °C for 30 s, and annealing at 63 °C for 30 s following an extension stage at 72 °C for 30 s. The amplification was ended with end extension stage at 72 °C for 10 min. Primer sequences are listed in Appendix A. 

### 2.3. Animal Slaughter

Nine female mice of the two groups (WT = 4 and *Zbed6*^+/−^ = 5) were fed until the age of 8 weeks and euthanized by cervical dislocation. The hindlimbs were removed carefully, and the combination of muscle tissue of its two hindlimbs was dissected separately and divided to provide two specimens. The skeletal muscle of left side was flash-frozen immediately and then transferred into −80 °C refrigerator for long-term storage for further testing in the future. The skeletal muscle of right side was first weighed as muscle mass in this study and then fixed with 4% paraformaldehyde solution (Solarbio, Beijing, China) for more than 24 h. Muscle mass and muscle weight ratio (muscle mass/body weight × 100%) was compared between *Zbed6*^+/−^ and WT groups. All muscle tissues with the same purpose were collected from the same place. 

### 2.4. Western Blot Analysis

Skeletal muscle was taken out from −80 °C refrigerator. Total proteins of muscle tissue were extracted using Minute TM Total Protein Extraction Kit for Animal Cultured Cells/Tissues (Invent Biotech, Plymouth, MN, USA). The proteins were used for concentration analysis under BCA Protein Assey Kit (Beyotime, Shanghai, China) and then subjected to Western blot analysis with the following antibodies: anti-ZBED6 antibody (HPA068807, 1:500, ATLAS, Stockholm, Sweden) and anti-Gapdh antibody (HRP-60004, 1:50,000, Proteintech, Chicago, IL, USA). The blots were developed using HRP-conjugated secondary antibodies. Picture were captured by imaging system (Taon 4600, Shanghai, China) and quantified by Image J software (Fiji, National Institutes of Health, Bethesda, MD, USA).

### 2.5. Histochemistry and Analysis of Myofiber 

The chemically fixed muscle was removed from 4% paraformaldehyde solution and then put in the dehydration box. The tissue in the dehydration box was put in dehydrator (Donatello, DIAPATH) to dehydrate with gradient alcohol (Ethanol, 100092683, Sinopharm Group Chemical Reagent Co., Ltd., Shanghai, China) and embedded in paraffin to make paraffin sections. The slides were created from cross-sections by a pathology slicer (RM2016, Shanghai Leica Instrument Co., Ltd., Shanghai, China) and then stained by hematoxylin and eosin (G1003, Servicebio, Wuhan, China). The whole slices were scanned and imaged by a slice digital scanner (Pannoramic 250FLASH, 3DHISTECH Ltd., Budapest, Hungary), the software CaseViewer 2.4 (3DHISTECH Ltd., Budapest, Hungary) was used to capture images of HE sections at 10× magnification, and Image J was applied for morphological observation. At least 20 myofibers were randomly selected from 3 fields per slice to detect the average area of the myofiber. 

### 2.6. RNA Isolation and Library Preparation

The total RNA of all muscle samples from the *Zbed6*^+/−^ and WT mice was isolated and purified using the RNeasy Mini kit (QIAGEN, Dusseldorf, Germany), in accordance with the operation protocol of the kit of the manufacturer. The quantity and purity of total RNA were then quality-controlled by NanoDrop ND-1000 (NanoDrop, Wilmington, DE, USA), and the integrity of RNA was checked by Bioanalyzer 2100 (Agilent, CA, USA). Only the RNA with concentrations >50 ng/μL, RIN value > 7.0, and total RNA > 1μg was submitted to downstream qPCR and RNA-seq analysis. cDNA library was then prepared from the qualified RNA using PrimeScript RT reagent kit (Takara Biomedical Technology Co., Ltd., Beijing, China) for Veriti 96 thermocycler (Applied Biosystems, San Francisco, CA, USA).

### 2.7. Primer Design and Quantitative Real-Time PCR

Quantitative real-time PCR (qRT-PCR) analysis was performed using ABI MicroAmp Optical 384-well reaction plates on an ABI 7900 real-time PCR instrument (Applied Biosystems) and Taq Pro Universal SYBR qPCR Master Mix (Q712, Vazyme Biotechnology Co., Ltd., Nanjing, China). The qRT-PCR protocol included 40 cycles of denaturation at 95 °C for 10 s and annealing and extension at 60 °C for 30 s after a pre-denaturation stage at 95 °C for 30 s. *Gapdh*, *β-Actin*, and *Rpl41* were used as the housekeeping gene to normalize all the results, which were presented as 2^−ΔΔCt^. The primers used were designed on the online website Primer3 (https://bioinfo.ut.ee/primer3-0.4.0/, accessed on 15 February 2023) and are listed in Appendix A.

### 2.8. RNA-Seq Analysis

Transcriptome paired-end sequencing of all RNA samples was conducted at LC Sciences (Hangzhou, China) using an Illumina NovaseqTM 6000 (LC Bio Technology CO., Ltd. Hangzhou, China) with PE150 sequencing mode following standard procedures. To get high-quality clean reads, reads were further filtered by Cutadapt (https://cutadapt.readthedocs.io/en/stable/, accessed on 15 February 2023, version: cutadapt-1.9). Then, the sequence quality was verified using FastQC (http://www.bioinformatics.babraham.ac.uk/projects/fastqc/, accessed on 15 February 2023, 0.11.9) including the Q20, Q30, and GC content of the clean data. The reference genome assembly of mouse (version GRCm39) was downloaded from Ensembl website (http://ftp.ensembl.org/pub/release-107/fasta/mus_musculus/dna/Mus_musculus.GRCm39.dna.toplevel.fa.gz, accessed on 15 February 2023). HISAT2 tools were used for mapping between the acquired clean reads and the reference genome. Gene expression of the muscle tissue of all mice was calculated by using featureCounts software. Moreover, differentially expressed analysis between *Zbed6*^+/−^ and WT mice was analyzed using the package DESeq2 of R software (version 4.2.0, http://cran.r-project.org/, accessed on 15 February 2023). In addition, parameter *p* value < 0.05 and log_2_ fold change absolute value ≥1 were used as a standard to carry out differential expression genes. TPM value, which stands for transcript per million, for all annotated transcripts and can be considered as a percentage of RPKM/FPKM values, was transferred from read counts in R software. All results of differential expression analysis were visualized by R. 

### 2.9. Enrichment Analysis 

Gene ontology (GO) enrichment analysis and Kyoto Encyclopedia of Genes and Genomes (KEGG) pathway enrichment analysis were performed in the DEGs between WT and *Zbed6*^+/−^ mice derived from differential expression analysis. The enrichment analysis was carried out on the online tool KOBAS-intelligence (http://bioinfo.org/kobas, accessed on 15 February 2023). *p* value ≤ 0.05 was used as the threshold to identify the significant GO terms and KEGG pathway using FDR correction method of Benjamini and Hochberg. 

### 2.10. Candidate Gene Selection

The DEGs of female mice between WT and *Zbed6*^+/−^ were overlapped with the target genes of ZBED6 acquired using chromatin immunoprecipitation (ChIP) followed by high-throughput DNA sequencing (ChIP-seq) in C2C12 cells that were published to extract for common genes. The candidate genes were chosen by querying the function of each common gene on NCBI (https://www.ncbi.nlm.nih.gov/, accessed on 15 February 2023). The ChIP-seq data of C2C12 cells were also provided by Leif Andersson (Science for Life Laboratory, Department of Medical Biochemistry and Microbiology, Uppsala University).

### 2.11. Statistical Analysis

All the results reported in this study were biologically replicated at least three times. The statistical analyses were conducted by GraphPad Prism 9.4.1 (GraphPad Software Inc., La Jolla, CA, USA). Data are presented as mean ± standard error of the mean (SEM). The *p* value was determined by Student’s t-test and indicated the significant level. *p* < 0.05 was considered statistically significant.

## 3. Results

### 3.1. Zbed6 Single-Allele Knockout Promotes the Growth of Skeletal Muscle 

The phenotypes of the WT and *Zbed6*^+/−^ mice were collected at the age of 8 weeks. Nine mice from two groups (n = 4:5) were slaughtered to measure their skeletal muscle. The procedure of sample collecting was followed as shown in Figure 1A, and the genotyping of mice was identified by PCR and subsequent gel electrophoresis. Bands of 596 bp represent WT mice, while bands of 561 bp represent *Zbed6*^+/−^ mice (Figure 1B). The Western blot analysis with an anti-ZBED6 antibody showed that a 43.4% decrease in the gray scale value of *Zbed6*^+/−^ compared to that of WT mice indicates that ZBED6 protein content significantly decreased in the skeletal muscle of *Zbed6*^+/−^ mice (Figure 1C,D).

The measurement of muscle growth showed that the muscle mass of *Zbed6*^+/−^ increased significantly compared to WT, which means the single-allele knockout of ZBED6 also could increase muscle weight in mice. In addition, the muscle mass of *Zbed6*^−/−^ also increased extremely significantly compared to either WT or *Zbed6*^+/−^ mice (Figure 2A and Appendix A), consistent with a previous study on *Zbed6*^−/−^ mice. In addition, *Zbed6*^+/−^ mice showed a significantly higher muscle weight ratio compared to WT mice (Figure 2B). 

To further investigate the effect of *Zbed6* single-allele knockout on mice muscle, the muscle tissue was paraffin-fixed for slicing. Hematoxylin and eosin (H&E) staining and its quantitative analysis clearly revealed that there was a significant increase in the area of myofiber in *Zbed6*^+/−^ mice compared with the WT group. The image and the quantitative analysis of the myofiber showed the same trend as the muscle mass, which showed a significant increase between the WT and *Zbed6*^+/−^ mice (Figure 2C,D and Appendix A). In addition, *Zbed6* single-allele knockout in mice can clearly lead to significant changes in skeletal muscle. 

### 3.2. Zbed6 Single-Allele Knockout Did Not Increase Igf2 mRNA Expression in Mice Muscle

In a previous study, the ZBED6–*IGF2* axis had a major effect on the muscle growth of placental mammals, such that we first detected the change in *Igf2* expression to reveal the effect of *Zbed6* single-allele knockout on *Igf2* in mice muscles. According to the results of qRT-PCR, with *Gapdh*, *β-Actin*, and *Rpl41* as internal references, there was no significant difference in the mRNA expression of *Igf2* in muscles between the WT and *Zbed6*^+/−^ groups (Figure 3A–C), which showed that *Igf2* was not the major effect gene to increase the muscle growth in *Zbed6*^+/−^ mice. 

### 3.3. RNA-Seq of WT and Zbed6^+/−^ Mice 

Transcriptome analysis of the muscle tissue of mice was conducted to explore the transcriptional changes aroused by *Zbed6* single-allele knockout. The pipeline was used to find DEGs between WT and *Zbed6*^+/−^ muscle tissues from the RNA-seq data (Figure 4A,B). The results of the quality control of transcriptional sequencing were placed in the attachment (Appendix A), showing a high quality for the RNA-seq data. 

The WT and *Zbed6*^+/−^ mice of each group showed a high degree of correlation (relevance value ≥ 0.97) (Figure 5A), which revealed the reliability of the RNA-seq data. All 55,414 genes were in the gene model, and we created statistics based on the TPM value of each sample (Appendix A). The TPM value was smaller than 1 for about 72.2–76.3% of more than 40,000 genes, the TPM value was between 1 and 10 for about 13.2–14.7% of the genes, and the value was more than 10 for about one-eighth of all the genes in each sample (Figure 5B).

There are 30,275 genes in the muscle expressed in both *Zbed6*^+/−^ and WT mice (Appendix A). According to the RNA-seq data, besides the similar expression of *Igf2*, we also found that there is no significant difference between WT and *Zbed6*^+/−^ mice by inspecting the value of TPM (Figure 3D), which was in agreement with the RT-qPCR result (Figure 3A–C). All these results for the expression of *Igf2* by qRT-PCR and RNA-seq suggested that *Igf2* was not the main effector gene for muscle increase in *Zbed6*^+/−^ mice. 

A differential expression analysis showed there were 663 differential expressed genes (DEGs) obtained from skeletal muscle, including 309 upregulated genes and 354 downregulated genes (Figure 6A,B and Appendix A), and the log2FC ratio ranged from −6.74 to 5.33 (Figure 6B). The hierarchical cluster analysis was performed for the DEGs by using ggplot2 in the R software package, which used a direct heatmap plot to visualize the result. This showed that the gene expression patterns of samples in *Zbed6*^+/−^ group were more similar for individuals in the WT group (Figure 6C), which also provided the reliability and accuracy of the data again. 

### 3.4. Enrichment Analysis of DEGs between ZBED6^+/−^ and WT Mice Muscle

Gene Ontology (GO) enrichment and Kyoto Encyclopedia Genes and Genomes (KEGG) enrichment were performed with the 663 DEGs. We received 2339 GO terms and 243 KEGG pathways in total, and 477 terms and 60 pathways were significantly enriched with *p* value ≤ 0.05 (Appendix A). The top 20 KEGG pathways were visualized in Figure 6E, such as type I diabetes mellitus, tuberculosis, and toll-like receptor signaling pathways. There were 38 terms of all the GO terms that corresponded to muscle, such as skeletal muscle contraction, muscle fiber development, muscle tissue development, skeletal muscle cell differentiation, structural constituent of muscle, etc., and two of them significantly corresponded to muscle growth under the condition of *p* value ≤ 0.05, including skeletal muscle contraction and the regulation of muscle contraction (Figure 6D and Appendix A). 

### 3.5. Zbed6^+/−^ May Increase the Muscle Growth of Mice via Upregulation of Barx2

The overlapping of the DEGs and ZBED6 target genes that arrived from murine C2C12 cells was performed for candidate gene searching [1]. In total, 28 common genes of 663 DEGs were found as candidate genes when overlapping with 1705 ZBED6 target genes (Figure 7A). There were 14 upregulated genes and 14 downregulated genes among the 28 candidate genes (Figure 7B,C). By scanning the functions of the 28 candidate genes one by one, we noticed *Barx2*, which is related to muscle growth and development, as identified in previous studies. The TPM value of the RNA-seq data showed a significant increase compared to WT mice (Figure 7D). On the basis of the individuals used for transcriptomic analysis, the number of individuals in the experimental group was increased, and the expression of the *Barx2* gene in the muscle of the *Zbed6*^+/−^ detected by qRT-PCR was also significantly increased. These results suggested that *Zbed6*^+/−^ may increase the muscle growth of mice via the upregulation of *Barx2*.

### 3.6. Verification of Gene Expression Profiles

To confirm the reliability of mRNA expression pattern, 10 DEGs were selected for validation using qRT-PCR under the criteria of *p* < 0.05 and logFC ≥ 1. *β-actin*, *Rpl4*, and *Gapdh* were selected as the reference genes, and it was assumed that the expression of the reference genes in all samples was constantly expressed. By comparing the DEGs’ expression levels, we found that the results were consistent with the upward and downward trends of the RNA-seq results, indicating that our data were reliable (Figure 8A,B).

## 4. Discussion

IGF2 is a crucial growth-promoting factor in postnatal growth and causes a major quantitative trait locus that affects muscle skeletal muscle growth and fat deposition [20]. The interaction of ZBED6 and *IGF2*, called the ZBED6–*IGF2* axis, plays an important role in the growing process. Several studies showed the effect of ZBED6 on the muscle growth of mice, pigs, and other mammals [2,9,10,21]. Conditional *Zbed6* knockout mice were successfully generated using homologous recombination. These mice were *Zbed6*-ablated in all tissues via being crossed with mice that expressed Cre in the germ line [2]. This study demonstrates that the single-allele knockout of *Zbed6* promotes muscle growth in mice; the changes are not primarily through the ZBED6–*Igf2* axis, rather some latent ZBED6 target genes affect muscle growth and development. Therefore, the extra regulating mechanism of ZBED6 functioning on muscle has been expanded through the *Zbed6*^+/−^ mice model. Thus, our study has revealed a new ZBED6 target gene involved in regulating the growth and development of mice skeletal muscle. 

What we noticed is that the single-allele knockout of *Zbed6* showed a significant promoting effect on skeletal muscle. The *Zbed6*^+/−^ mice exhibited greater muscle growth at the age of 8 weeks, the age at which mice just reach sexual maturation, compared to WT mice (Figure 2A). At the same time, the *Zbed6*^−/−^ mice showed an extremely significant increase in muscle mass, which was in accord with the significant growth of *Zbed6*^−/−^ mice when compared with WT mice, as reported in a previous study [2]. An analysis of the H&E staining found there was consistency with the results on muscle mass and muscle ratio, which may provide a reasonable explanation: the muscle mass of *Zbed6*^+/−^ mice increased due to the increase in the average area of the myofiber (Figure 2B). The observed phenomenon is that the muscle growth showed a significant dose effect in WT, *Zbed6*^+/−^, and *Zbed6*^−/−^ mice, even though the mechanism leading to this dose effect is not yet clear. 

Several previous studies confirmed that *IGF2* is the capital gene for increasing muscle mass in *ZBED6*^−/−^ pigs and mice [2,9,10]. The knockout of *Zbed6* and the knockin of *Igf2* in mice resulted in an increased circulating IGF2 level and an increased growth rate compared to WT littermates [2]. The knockout of *ZBED6* in pigs resulted in a three-fold higher *IGF2* expression level in muscle and an eight-fold higher *IGF2* expression level in serum. Thus, we took the lead in checking the mRNA expression of *Igf2* in muscle by qRT-PCR and RNA-seq analysis in this experiment. The mRNA expression of the *Igf2* of muscle has no significant difference between *Zbed6*^+/−^ and WT mice, suggesting that the effect of single-allele knockout of *Zbed6* on muscle growth and development is mediated through targets other than *Igf2*, which also means that *Zbed6*^+/−^ and *Zbed6*^−/−^ possess different mechanisms for regulating muscle growth in mice. 

To figure out what really functions in the process, during which the muscle of *Zbed6*^+/−^ mice grew faster than that of WT mice, muscle transcriptome analysis was performed by searching for DEGs between the two groups. The gene expression profiles of the muscle of *Zbed6*^+/−^ and WT mice are different. Although the PCA result showed that it cannot be separated into two entirely different groups due to some exceptions away from the group (Figure 6A), the cluster analysis of the DEGs indicated that the data were accurate and dependable and could satisfy the subsequent difference analysis (Figure 6C). The 633 genes were scanned as DEGs between *Zbed6*^+/−^ and WT. Compared to the great changes in the expression profile of the thousands of genes between *Zbed6*^−/−^ and WT mice, from the research of Younis et al. [2], the amount of DEGs between *Zbed6*^+/−^ and WT are decreased. *Igf2* is no longer one of the DEGs that is in accord with the level of mRNA expression. The DEGs are enriched to 40 GO terms that are related to muscle growth and development, and two of them are significantly related.

The results of the ChIP-seq analysis already investigated that ZBED6 binds thousands of sites in the C2C12 cells and successfully annotated 1705 genes as ZBED6 target genes [1]. There are 28 overlapping genes, with 663 DEGs and 1705 ZBED6 target genes. *Barx2*, one of these common genes, is a homeobox gene of the Bar class that is abundantly expressed in skeletal muscle cells [22] and was chosen as a candidate target gene of ZBED6. Barx2 is a key intrinsic regulator of myogenesis and seemed to be a novel marker gene for muscle progenitor cells and an important regulator of muscle growth and repair [23,24]. BARX2 is expressed in skeletal muscle and is upregulated during differentiation of the skeletal myotube [25]. A previous study showed that the myotube formation was prevented by the inhibition of *Barx2* expression but was accelerated by the overexpression of *Barx2* in primary limb bud cell culture [26]. *Barx2* also promotes myotube formation in C2C12 myoblasts by the stable transfection of a pcBarx2 expression plasmid [26]. Mice that lack the *Barx2* gene (*Barx2*^−/−^) show muscle growth, muscle atrophy, and defective muscle repair postnatally [27]. The satellite cells and myoblasts derived from *Barx2*^−/−^ mice show a decline in proliferation and differentiation ability and a decrease in the expression of differentiation related factors, respectively [27]. All these results indicate that *Barx2* plays an important role in promoting the growth and differentiation of skeletal muscle. Here, the mRNA expression of *Barx2* in *Zbed6*^+/−^ mice was raised significantly compared to that of WT mice, confirming that *Barx2* may play an important role in upregulating the growth of muscle when single-allele knockout of *Zbed6* occurs. Our results first reported the effect of *Zbed6* single-allele knockout via *Barx2*, a target gene other than *Igf2*, which promotes the growth and development of muscle in female mice. Further study could identify the mechanism of the ZBED6–*Barx2* axis on skeletal muscle growth in mice.

## 5. Conclusions

In conclusion, our study described the effect of muscle when *Zbed6* single-allele knockout provided the transcriptional data of muscle from *Zbed6*^+/−^ and WT mice at the age of 8 weeks, demonstrating that *Zbed6*^+/−^ may promote the growth and development of mice via the upregulation of *Barx2.* This study may contribute to the further exploration of the other target genes of ZBED6 that have not yet been studied. 

## Figures and Tables

**Figure 1 biology-12-00325-f001:**
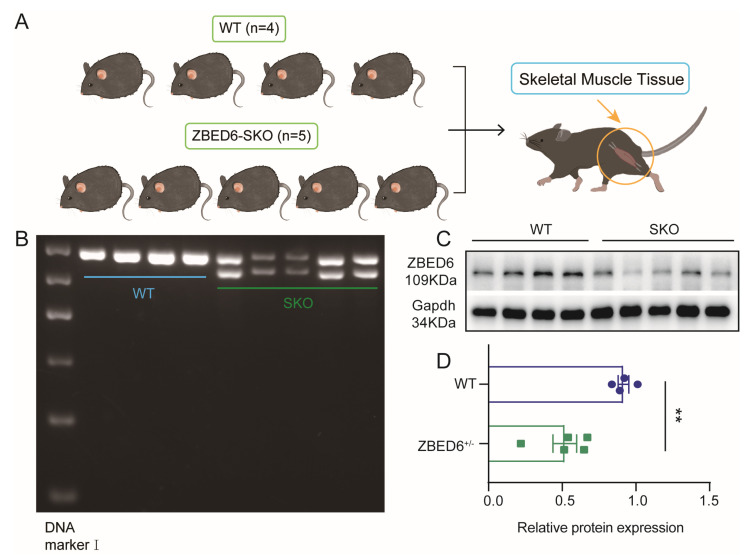
Schematic overview of the production of mice in this study. (**A**) Skeletal muscle tissues were weighed and collected from WT and *Zbed6*^+/−^ mice. (**B**) Genotyping of the different genotype mice was identified by polymerase chain reaction (PCR). The 596 bp band represents the WT mice, and the 561 bp band represents the *Zbed6*^+/−^. (**C**,**D**) Western blot analysis of ZBED6 in skeletal muscle from *Zbed6*^+/−^ mice and WT mice. The results are expressed by the means ± SEMs. WT represents wild-type mice, and SKO represents *Zbed6*^+/−^ mice. ** *p* < 0.01, Student’s t-test. Points in different colors represent actual data of mice.

**Figure 2 biology-12-00325-f002:**
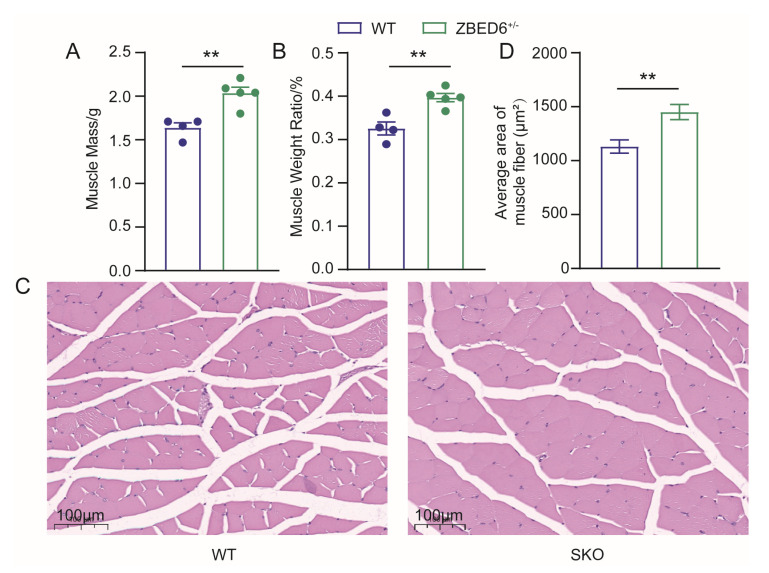
Phenotypes of slaughtered WT and *Zbed6*^+/−^ mice. (**A**,**B**) Dissected muscle mass and muscle weight ratio of WT and *Zbed6*^+/−^ mice. Points in different colors represent actual data of muscle traits. (**C**) H&E staining of the skeletal muscle of WT and *Zbed6*^+/−^ mice. SKO represents *Zbed6*^+/−^ mice. Bar, 100 μm. (**D**) Differences in the average area of muscle fiber quantitative analysis in mice. The results were expressed by the means ± SEMs. ** *p* < 0.01, Student’s t-test. Points in different colors represent actual data of mice.

**Figure 3 biology-12-00325-f003:**
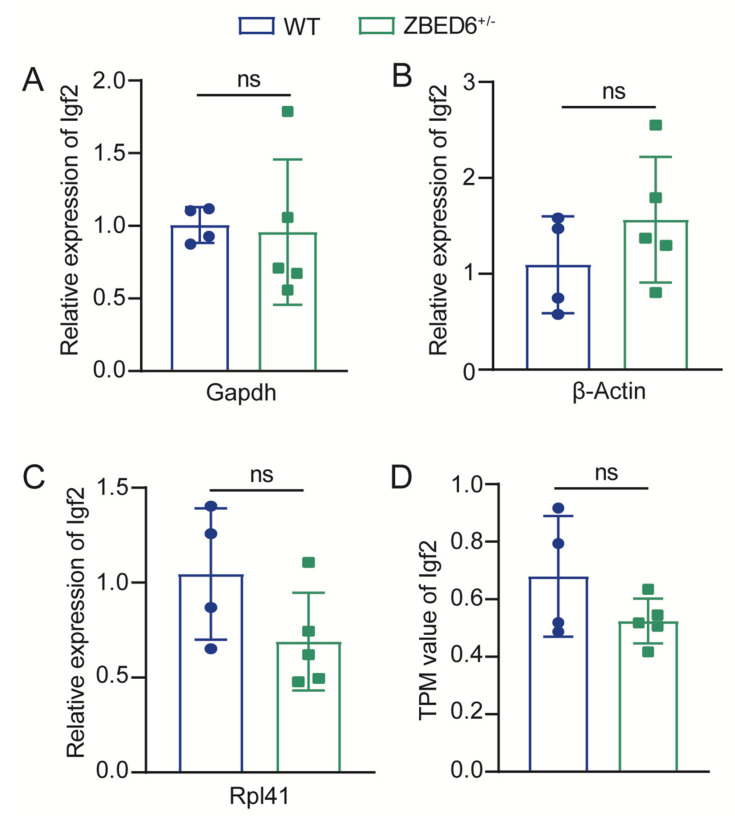
*Igf2* level shows no significant change between WT (n = 4) and *Zbed6*^+/−^ (n = 5) mice skeletal muscle. (**A**–**C**) qRT-PCR results of *Igf2* mRNA expression in mice skeletal muscle from WT and *Zbed6*^+/−^ mice using *Gapdh*, *β*-*Actin*, and *Rpl41* as internal reference genes. (**D**) TPM value of *Igf2* expression levels identified from RNA-seq data in mice skeletal muscle. The results were expressed by the means ± SEMs, ns represents not significant, Student’s *t* test. Points in different colors represent actual data of mice.

**Figure 4 biology-12-00325-f004:**
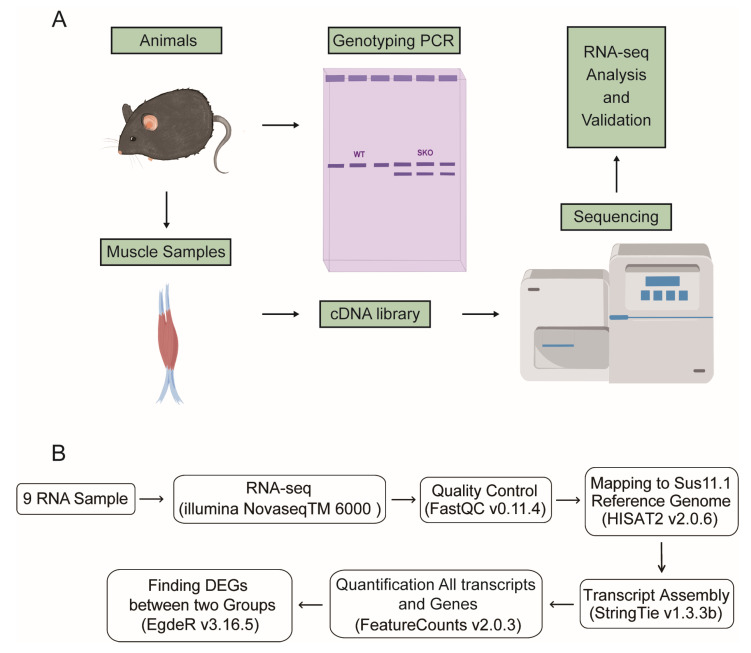
Schematic diagram of technical route. (**A**) The sample processing process includes two parts: the first part is genotyping, and the second part is muscle tissue RNA extraction, library construction, and sequencing. (**B**) Flowchart of RNA-sequencing and transcriptome analysis in this study.

**Figure 5 biology-12-00325-f005:**
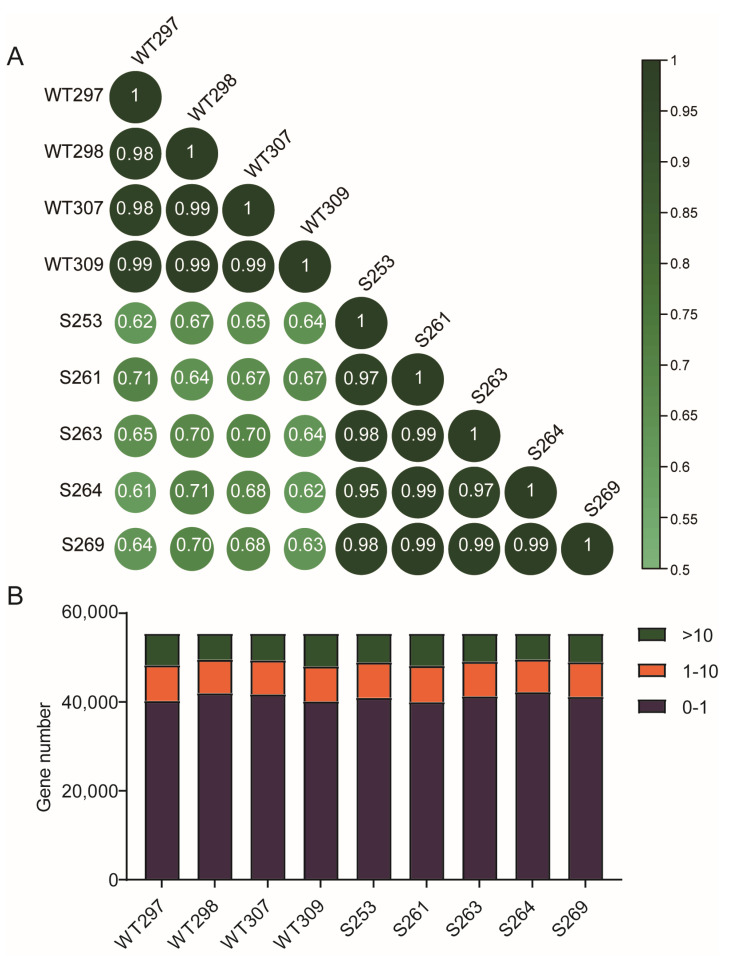
Description of RNA-sequencing of all muscle samples from WT and *Zbed6*^+/−^ mice. (**A**) The correlation analysis of WT and *Zbed6*^+/−^ groups. The larger the value is, the stronger the correlation is. (**B**) The statistic chart of TPM value of all skeletal muscle samples.

**Figure 6 biology-12-00325-f006:**
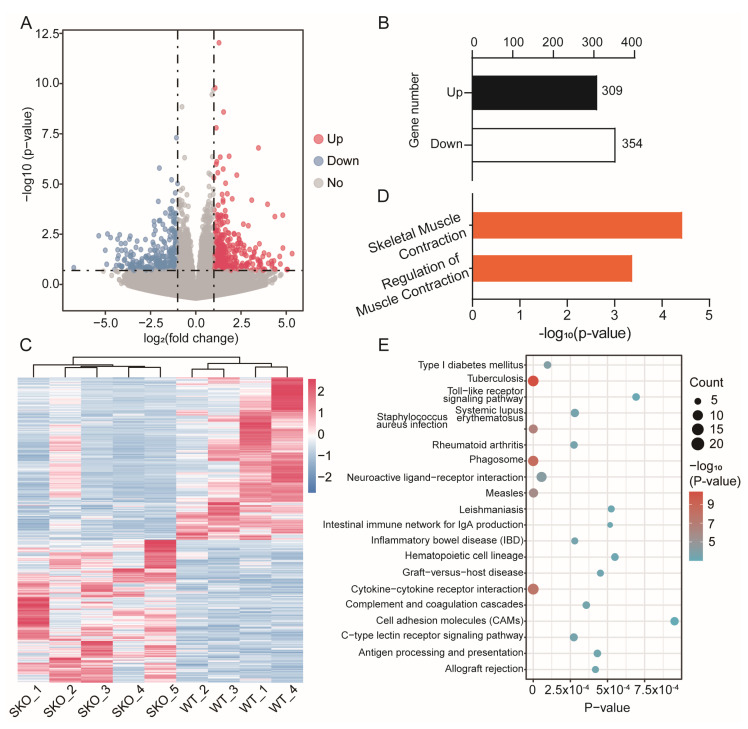
Statistical analysis of the expressed genes. (**A**) The volcano plot of all DEGs between muscle of *Zbed6*^+/−^ and WT mice. Red spots represent upregulated DEGs, and blue spots represent downregulated DEGs. The gray spots indicate genes with no significant difference in expression. The horizonal black dotted line shows that the *p* value is equal to 0.05, while the vertical dotted line shows that log2FC is equal to −1 (left) or more than 1 (right). (**B**) Statistics of up- and down-regulating DEGs. (**C**) Hierarchical clustering of DEGs. Red indicates upregulated DEGs, and blue indicates upregulated DEGs. The darker the color is, the more changes in gene expression there are. (**D**) Significantly enriched Gene Ontology terms of DEGs using *p* value equal to 0.05 as threshold. (**E**) Top 20 KEGG pathways of DEGs.

**Figure 7 biology-12-00325-f007:**
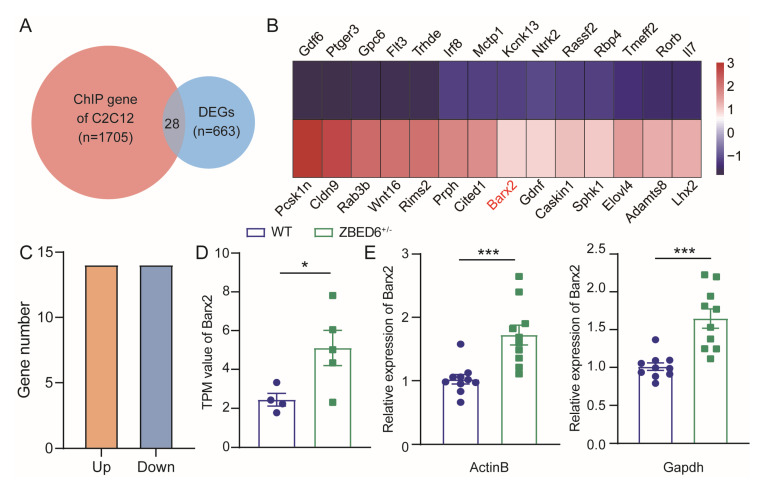
The identification of candidate genes. (**A**) Overlapped DEGs and ChIP genes. Red circle indicates ZBED6 target genes derived from ChIP-seq of murine C2C12 cells. Blue circle indicates DEGs. (**B**) Heatmap of common 28 genes. The genes marked in red and blue were upregulated and downregulated, respectively. (**C**) Number of upregulated and downregulated genes. (**D**) TPM value of *Barx2*, n = 4:5. (**E**) The mRNA expression level of *Barx2*, n = 10:10. The results were expressed by the means ± SEMs. * *p* < 0.05, and *** *p* < 0.001; Student’s t-test. Points in different colors represent actual data of mice.

**Figure 8 biology-12-00325-f008:**
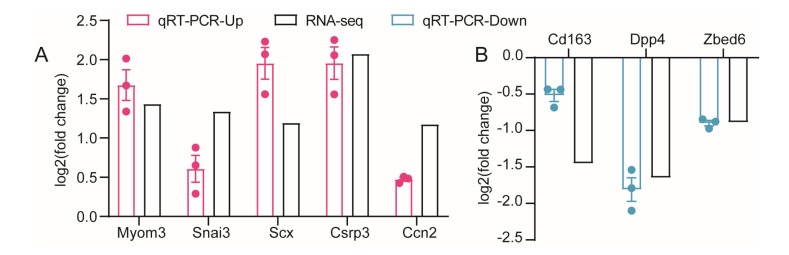
The verification of DEGs profile. (**A**,**B**) The verification of expression levels of 5 up-regulated DEGs and 3 downregulated DEGs on the basis of qRT-PCR and RNA-seq data. Log2 (fold change) > 0 or <0 indicates the DEG was upregulated or downregulated in *Zbed6*^+/−^ group compared with WT group. Points in different colors represent actual data of mice.

## Data Availability

The raw sequence data reported in this paper were deposited into the CNGB Sequence Archive (CNSA) of the China National GeneBank DataBase (CNGBdb) with accession number CNP0003693, which are publicly accessible at https://www.cngb.org/, accessed on 14 February 2023.

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
