# Peer review of "Effect of Zbed6 Single-Allele Knockout on the Growth and Development of Skeletal Muscle in Mice"

_biology, 2023, doi:10.3390/biology12020325_

Round 1

Reviewer 1 Report

The manuscript titled “Effect of ZBED6 Single Allele Knock-out on the Growth and Development of Skeletal Muscle in Mice” compared muscle mass and the mRNA expression profile in the hindlimb muscle between ZBED6 Single-allele Knock-out (Zbed6+/-) mouse and wild type (WT), overlapped the DEGs and the ZBED6 target genes from the ChIP-seq data in C2C12 cell, presented that Zbed6+/- may increase the muscle growth of mice via up-regulation of Barx2. This experiment features original topics, a logical design, appropriate analysis methods. However, some results are needed to be changed and the manuscript is needed to be checked and corrected thoroughly before publication.

Major concern:

Twelve female mice of the three groups (WT = 4, Zbed6+/- = 5, Zbed6-/- = 3) were used in current study. Differing from other results, there were just 3 WT and 4 Zbed6+/- in the qPCR results of Figure 7E. The authors should use the qPCR results from all individuals. Also, the expression of Barx2 gene in muscle of Zbed6+/- and WT should be detected by western blotting.

The authors proposed that Zbed6+/- may increase the muscle growth of mice via up-regulation of Barx2. So, the presentation in the Conclusion section (Line 403-404) should be changed.

Minor comments:

1.     The gene symbols are needed to be italic and the protein symbols are needed to be in upper case.

2.     Line 17, studied proved?

3.     L54, change “intro 3” to “intron 3”

4.     There should be spaces between numbers and units.

5.     L204 and L210, the length of PCR products of different genotypes in Results section is not consist with it in Figure 1.

6.     In Figure 2, please explain abbreviation for WT, SKO and KO.

7.     Numbers which more than 4 digits should be comma-separated every 3 digits.

8.     L309, remove “Transcriptome”

Reviewer 2 Report

Liu and collaborators demonstrated, using a Zbed6 single allele knockout mice model, that knockout mice have higher muscle area and mass, and the authors purposed a mechanism of muscle growth independent of Igf2, via Barx2 gene. Although informative for skeletal muscle researchers, limitations/comments are addressed below.  In summary, in my opinion, this is valuable work, but some experiments are lacking to confirm some important conclusions.

1.     Why single-allele strategy should be tested instead of a double knockout? The authors should include a sentence in the introduction describing the importance of this strategy.

2.     Why authors chose female mice?

3.     The authors should be more specific regarding the hindlimbs muscle evaluated. It is not possible to know only by the description given. It was soleus? Gastrocnemius? EDL? A combination?

4.       Are the CHIP-seq data publicly available? If yes, a reference number is required.

5.     Authors should evaluate gene expression and protein levels of ZBED6 in each group to confirm the gene knockout. Moreover, in Supplementary Table 8, the gene Zbed6 is not differentially expressed, the authors should include and discuss this finding, as it is expected Zbed6 to be downregulated in the knockout group.

6.       Supplementary Tables 7 and 8 are demonstrating the logFC values for the WT group. The opposite of what the authors demonstrated in the manuscript. For example, the Barx2 gene has a logFC = -1.03385. The authors should correct this.

7.       It is not clear the criteria used to select 10 DEGs for qRT-PCR analysis. Why did the authors not include Zbed6 expression quantification in qPCR?

8.       The author did not confirm the causal relationship between the increase of Barx2 and muscle hypertrophy. I wonder if Barx2 treatment by recombinant protein in C2C12 myotubes may help to confirm the causality. I suggest authors include some additional data/experiments regarding this point. 

Round 2

Reviewer 2 Report

The authors addressed the mentioned comments.